# Removal of Transition Metals from Contaminated Aquifers by PRB Technology: Performance Comparison among Reactive Materials

**DOI:** 10.3390/ijerph18116075

**Published:** 2021-06-04

**Authors:** Celia Margarita Mayacela-Rojas, Antonio Molinari, José Luis Cortina, Oriol Gibert, Carlos Ayora, Adalgisa Tavolaro, María Fernanda Rivera-Velásquez, Carmine Fallico

**Affiliations:** 1Faculty of Civil and Mechanical Engineering, Universidad Técnica de Ambato, Ambato 180103, Ecuador; 2Department of Civil Engineering, Università della Calabria, 87036 Rende, Italy; ant.molinari2002@libero.it (A.M.); carmine.fallico@unical.it (C.F.); 3Barcelona Research Center for Multiscale Science and Engineering, UPC-BarcelonaTECH, C/Eduard Maristany, 10–14 Campus Diagonal-Besòs, 08930 Barcelona, Spain; jose.luis.cortina@upc.edu (J.L.C.); oriol.gibert@upc.edu (O.G.); 4Departament de Geociències, Institut de Diagnosi Ambiental i Estudis de l’Aigua (IDÆA-CSIC), c/Jordi Girona 18 UPC Campus Norte, 08034 Barcelona, Spain; cayora@ija.csic.es; 5Institute on Membrane Technology, National Research Council (C.N.R.-I.T.M.), University of Calabria, 87036 Rende, Italy; a.tavolaro@itm.cnr.it; 6Alternative Energies and Environment Research Group, Escuela Superior Politécnica de Chimborazo, Panamericana Sur km 1 1/2, Riobamba 060101, Ecuador; mariaf.rivera@espoch.edu.ec

**Keywords:** aquifers remediation, permeable reactive barriers, transition metals, vegetable fibers, zeolites, bach tests, column tests

## Abstract

The most common reactive material used for the construction of a permeable reactive barrier (PRB) is zero valent iron (ZVI), however, its processing can generate corrosive effects that reduce the efficiency of the barrier. The present study makes a major contribution to understanding new reactive materials as natural and synthetic, easy to obtain, economical and environmentally friendly as possible substitutes for the traditional ZHV to be used as filters in the removal of three transition metals (Zn, Cu, Cd). To assess the ability to remove these pollutants, a series of batch and column tests were carried out at laboratory scale with these materials. Through BACH tests, four of seven substances with a removal percentage higher than 99% were prioritized (cabuya, natural clinoptilolite zeolites, sodium mordenite and mordenite). From this group of substances, column tests were performed where it is evidenced that cabuya fiber presents the lowest absorption time (≈189 h) while natural zeolite mordenite shows the highest time (≈833 h). The latter being the best option for the PRB design. The experimental values were also reproduced by the RETRASO code; through this program, the trend between the observed and simulated values with respect to the best reactive substance was corroborated.

## 1. Introduction

Most of the world’s water needs are satisfied by groundwater, which makes up about 96% of the entire water supply of our planet [1] and which is often the only source of fresh water supply, especially in arid and semi-arid regions. The interaction with groundwater generated by multiple anthropogenic activities, especially agricultural and industrial, often produces pollution and degradation of its quality, which in many cases leads to a high risk for human health. In this context, the presence in groundwater of transition metals is particularly dangerous. Transition metal contamination is mainly produced by industrial activities, as well as by numerous other forms of interaction between anthropic activities and the environment (waste production, combustion of hydrocarbons, vehicular traffic, use of pesticides and others). Transition metals are toxic and/or carcinogenic species, with a high level of risk for human health, hence it is of fundamental importance to ensure the preservation of the good quality of this precious resource, which nowadays is subjected to an increasing use [2,3,4,5,6]. During the last decade, ex-situ remediation technologies (pump and treat) decreased in favor of in-situ remediation methods (PRBs, bioremediation, thermal remediation) [7]. Among the most innovative and promising technologies used for the remediation of contaminated aquifers, the use of permeable reactive barriers (PRBs) is particularly advantageous [8,9,10,11,12]. This technology, which foresees that the contaminated groundwater passes through a barrier of reactive material inserted into the aquifer to intercept and treat the water flow, is particularly convenient, especially compared to the pump and treat system, as it constitutes a passive system that does not require energy and reduces the number of operations needed and hence the resulting overall costs [13,14,15]. By applying this technology, a key aspect is the proper choice of the reactive material employed to fill the PRB, which depends mainly on the type of contaminant that must be treated [16]. Zero-valent iron, (ZVI) is the reagent medium commonly employed in real-scale applications in PRBs to remediate contaminated sites, both at the micro- and macroscale [17,18,19,20,21,22]. However, new alternatives such as natural fibers, zeolites, among others, have gained interest among the scientific community as cheap and easy to obtain filtering material. In technical literature there is a large number of materials with high removal rates for different types of compounds. For example: activated sludge, basalt dust, biochars (wheat straw + coconut shell), blast furnace slag, bone char, cabuya, calcite, clinoptilolite, fly ash, granular activated carbon (GAC), maize cob, mulch and graveli, natural pyrited, non-living biomass, paper ash, peat, phosphatic compounds, plant shell and weed, recycled concrete, red mud, sand, sawdust + sand, sediments (SRBs, silica sand, limestone, compost) + (chicken manure) + (oak leaf, manure), steel slag dust, tree leaves, volcanic slug and pumice, waste foundry sand, woodchips, zeolites, zero valent iron (ZVI) [23,24,25,26,27,28,29,30,31,32,33,34,35,36,37,38,39,40,41,42,43,44].

For what concerns PRB construction, only a few PRBs are in operation around the world, mainly concentrated in advanced countries. The implementation of PRBs using local reagent medium could become a good option to reduce the total cost of installation and operation. The implementation of a PRB initially requires a strong baseline research, at laboratory scale, covering all key aspects of design and operation in order to avoid any mishaps at full scale.

The main objective of this study is to determine the removal capacity for three transition metals, Zn(II), Cu(II) and Cd(II), using some types of natural and synthetic reactive substances. A natural fiber, known as cabuya (*Furcraea andina*), two natural zeolites rich in clinoptilolite and mordenite, respectively, a synthetic zeolite (zeolite 4A), a sample of natural calcareous limestone from Ecuador and ZVI were used as reagent medium. These materials were selected on the basis of previous studies, which highlighted their large reactivity towards the above transition metals, compared to other investigated materials [45,46,47]. We: (1) determined porosity and analyzed grain size, (2) used batch tests to determine the adsorption capacity, (3) carried out hydraulic characterization, (4) evaluated the best reagent substances for column test. After this experimental phase, finally, (5) we used the reactive solids transport numbering code (RETRASO) [48] to simulate the column tests and compare experimental with simulated results.

The main results show that by means of the BACH tests, from a set of seven reactive substances initially considered, the most reactive substances were identified and subsequently divided into two groups according to their percentage of removal (Group 1 ≥ 99% and Group 2 < 98%). Under this criterion, four substances corresponding to group 1, were prioritized for column tests (cabuya, natural clinoptilolite zeolites, sodium mordenite and mordenite). In the following, column tests were performed to identify the relative adsorption capacity of each reagent medium with respect to each contaminant considered, and additionally, the times necessary to reach adsorption and saturation of the medium were obtained. The results show that: cabuya fiber presented the minimum absorption time (≈189 h) while natural zeolite mordenite shows a maximum absorption time (≈833 h). The latter being the best option for the PRB design. Subsequently, the sizing of the PRB was also carried out with the use of the mathematical code RETRASO, simulating the column tests for the four reactive substances considered (cabuya fiber, natural zeolites clinoptilolite, mordenite and sodium mordenite) and assuming a column length equal to 1 m, intermediate value between those obtained on the basis of the experimental values for the thickness of the PRB, the results obtained with the RETRASO code highlighted the lower reactivity of the cabuya fiber compared to the zeolitic substances considered.

## 2. Materials

The reactive materials considered in the present investigation were extensively characterized as described elsewhere [46,47].

### 2.1. Vegetable Fiber

The vegetable fiber considered is the *Furcraea andina*, also called fique fiber or, more commonly, cabuya. This belongs to the Agave family and is native to the Andean regions, although it grows also in many areas of Africa and Asia. Cabuya fibers have a typical polymeric structure, containing a high amount of cellulose and hemicellulose, and also of lignin, which has a high transition metal adsorption capacity. X-ray diffraction analysis highlighted the presence of relevant amount of cellulose and lignin [47], also characterized by Fourier transform infrared spectroscopy in attenuated total reflectance (FTIR-ATR). Cabuya fibers show also an adequate density, specific surface and diameter as reactive sorbent. The density of the vegetable fibers is very important, since it affects directly the hydraulic conductivity (*k*) value of the fibrous mass constituting the PRB, determining its actual hydraulic effectiveness. In fact, the *k* value of the PRB must always be greater than that of the contaminated aquifer, so that the water flow actually passes through the PRB. In the present investigation the decrease in hydraulic conductivity with increasing density of cabuya fibers was also verified, as done in a previous study focused on broom fibers [45].

### 2.2. Zeolite Samples

The natural zeolites are hydrated aluminosilicate minerals with a porous structure. These have commonly cation-exchange capacity, molecular sieving properties, catalytic role, sorption capacity. This material may have different chemical composition depending on the formation environment. The behavior and the adsorption performances can vary depending on several factors such as Si/Al ratio, cation type, number and location of adsorption sites. In the present study two natural zeolites were considered, the clinoptilolite, characterized by the Si/Al ratio equal to 1.6, and the mordenite, with a Si/Al ratio equal to 2.2. Furthermore, for these two natural zeolites, the elemental chemical composition analysis was carried out, using the inductively couple plasma mass spectrometry (ICP-MS) which showed that, both clinoptilolite and mordenite, are mainly composed by silicon, aluminum, calcium, sodium, iron and magnesium [47]. Both these zeolites were considered in the granular form. The corresponding grain size analysis showed for clinoptilolite and mordenite an amount of total retained material at a sieve with a size of 1.4 mm, respectively, equal to 61% and 64% of the total material. Since the size of the remaining percentages of these materials have grain sizes greater than 2 mm, both mediums were considered as a very coarse sand. We noted that the particle size characteristics of these materials influence the value of the porosity, which, like the density for the vegetable fibers, affects the value of the hydraulic conductivity (*k*) [45,49]. In addition to these two materials, another medium, was considered, the sodium mordenite zeolite. To derive this material, the mordenite natural zeolite was modified to obtain the sodium form, using an adaptation of the method reported by Jiménez-Cedillo [50]. A synthetic zeolite, zeolite 4A was used and its properties is described elsewhere Molinari et al. [47].

### 2.3. Calcareous Limestone

The mineralogical composition of the calcareous limestone reveals a mixture of well-crystallized concrete and quartz materials [47]. The structure of this material has commonly a high porosity and a large surface area. The analysis of the elemental chemical composition was carried out by ICP-MS, after total dissolution in nitric acid, indicating the presence of magnesium, calcium, barium, iron, silicon, aluminum, sodium and titanium. The Mg/Ca ratio is equal to 1.73 [47]. This material was also taken in the granular state and the corresponding grain size analysis showed that about 87% of the total material was retained on the sieve of 2.36 mm, while the remaining 13% was retained by the sieve of 1.40 mm.

### 2.4. Zero Valent Iron (ZVI)

The ZVI was characterized by scanning electron microscope (SEM) analysis, highlighting the presence of large specific adsorption surfaces [21,47]. The size of the particles used in the ZVI batch tests were 1.40 mm.

## 3. Methods

### 3.1. Porosity Determination

The total porosity (𝑛), defined as the ratio between the volume of the voids and the total volume of the sample investigated, was measured utilizing the laboratory densimetric method by the following equation [51,52]:(1)$$n=1-\frac{\rho_{bulk}}{\rho_{grain}}$$
where $\rho_{bulk}$ is the bulk mass density (kg/m^3^) and *𝜌_grain_* is the particle mass density (kg/m^3^).

The effective porosity (𝑛_𝑒_) is considered as saturated water content minus residual water content, namely:(2)$$n_{e}=n-\frac{V_{w}}{V}$$
where 𝑉 is the total volume (m^3^) and 𝑉_𝑤_ the water residual volume not drained by gravity (m^3^) [53], and it was measured under equilibrium conditions at 33 kPa of suction [54,55]. The effective porosity was also determined by the tracking method, namely by inserting a tracer (NaCl = 0.1 M) at the entrance of the column in which the material was placed and determining the average residence time of the tracer, using the following relationship:(3)$$n_{e}=\frac{Q}{\pi r^{2}h}\;\cdot \;\frac{\int_{0}^{\infty }tC\left(t\right)dt}{\int_{0}^{\infty }C\left(t\right)dt}$$
where the term $\frac{\int_{0}^{\infty }tC\left(t\right)dt}{\int_{0}^{\infty }C\left(t\right)dt}$ represents the average residence time (s), *Q* the flow rate in the examined column (m^3^/s), *r* and *h* represent the radius and height of the sample, respectively, (m) and *C*(*t*) the tracer concentration (kg/m^3^) considered at the generic *t* time [56,57].

### 3.2. Grain Size Analysis

For granular materials investigated (synthetic zeolite 4A, calcareous limestone, mordenite natural zeolite and clinoptilolite natural zeolite) the granulometric analysis was carried out by sieving. These materials, after being kept in an oven at 120 °C for 24 h, were passed through a series of sieves arranged one on top of the other, with decreasing openings net going from top to bottom (with steps 4.75, 2.36, 1.18, 0.60, 0.30, 0.15, 0.075 mm). Subsequently, the weightings of the solid fractions retained from each sieve were carried out, obtaining the respective granulometric curves.

### 3.3. Hydraulic Characterization Tests

The hydraulic conductivity of the reactive materials considered was determined in laboratory by flow cells used as constant head permeameters. A Mariotte bottle, which serves to fix the head, was connected by a tygon tube to the bottom of a cylindrical flow cell, with Plexiglas walls (internal diameter *d* = 6.4 cm and *h* = 15.3 cm) where the investigated material was placed. With these arrangements the water was pumped from the bottom to the top of the cell to allow the air escaping. Furthermore, two porous membranes were placed at each end of the cell to hold solid materials. The upper part of the cell was connected to the outside by a tygon tube from which the fluid came out [45]. Once the hydraulic head was fixed with the Mariotte bottle, a valve was opened to allow water filtering through the sample present into the cell-permeameter. By reading, on an appropriate graduated scale at fixed times, the water level in the Mariotte bottle, whose dimensions were known, it was possible to determine the volume of filtered water by means of the following relation:(4)$$v=\frac{\pi d^{2}}{4}\cdot h$$
where *d* is the diameter of the Mariotte bottle (m) and *h* the difference of water levels (m) for the interval times considered. Once determined the volume of filtered water, it was easy to define the water flow rate leaving the cell (*Q*). Knowing *Q*, it was possible to determine the filtration velocity in the porous medium:(5)$$v=\frac{Q}{\frac{\pi D^{2}}{4}}$$
where *D* is the diameter of the porous medium sample inside the cell-permeameter (m). Known the hydraulic head difference (Δ*H*) (m) and the piezometric gradient *J* = Δ*H*/*L* (--), from Darcy’s law it was possible to determine *k* (m/s), that is:(6)$$k=\frac{v}{J}$$

### 3.4. Batch Tests

Batch tests were carried out for a preliminary evaluation of adsorption capacity of each investigated reactive material medium [46]. The pH of each sample was measured, immediately after phase separation, by the use of a pH-meter (CRIPSON, GLP22). All measurements were performed with the use of an orbital mechanical agitator, at room temperature (20 ± 1 °C). The collected samples were filtered using 0.45 μm disc filters, acidified with concentrated nitric acid and stored in plastic bottles at 4 °C, before determining the concentrations of the pollutant. Initial concentration of transition metals investigated were, respectively, set to 100 mg/L for Zn, 50 mg/L for Cu and 40 mg/L for Cd. A solution volume of 0.04 L was assumed for all tests, the maximum contact time was set to approximately 30 h and the analysis were carried out considering the following contact times: 0.5–30 h, for all the reactive materials considered. Due to the interaction between the investigated medium and the contaminated solution (with a known concentration) an adsorption equilibrium was achieved. The amount of solute adsorbed (*q*) to the solid phase can be estimated as the difference between the initial concentration and the equilibrium concentration in solution after a fixed time.

#### Batch Adsorption Kinetics

Adsorption kinetics was evaluated by batch test using similar experimental conditions studying the variation of the metal ions concentration as a function of the contact time. For this purpose, the reactive materials adsorption vales as a function time were determined. The sorption kinetics was evaluated by the first order of Lagergren equation described by Equation (7) [58]:(7)$$\frac{dq_{t}}{dt}=K_{1}\left(q_{e}-q_{t}\right)$$
where $q_{t}$(--) is the mass of metal ions adsorbed in a time *t* (s), $q_{e}$(--) is that adsorbed mass at equilibrium and $K_{1}$(1/s) is the kinetic constant.

### 3.5. Metal Ions Column Adsorption Tests

The adsorption process can be highlighted by the adsorption isotherms, which represent the balance between the amount of adsorbed solute and the adsorbent mass unit. The known isotherms of Langmuir and Freundlich were considered in the present study. The Langmuir model is expressed by the Equation (8):(8)$$q_{e}=q_{m}\frac{k_{L}C_{e}}{1+k_{L}C_{e}}$$
where $q_{e}$ is the quantity of metal ions adsorbed per unit mass of adsorbent solid in equilibrium (--), *C_e_* is the concentration of the metal ion in the liquid phase in equilibrium (kg/m^3^), $q_{m}$ is the maximum adsorption capacity (--) and *K_L_* is the constant of adsorption (m^3^/kg) [59]. The Freundlich model is represented by Equation (9):(9)qe=kFCe1n
where *q_e_* is the amount of metal ion adsorbed per unit of solid mass in equilibrium (--), *C_e_* is the concentration of metal ions in the solution in equilibrium (kg/m^3^), *K_F_* is the constant of Freundlich (m^3^/kg) related to the binding energy, 1/*n* it is Freundlich’s heterogeneity factor and *n* is the degree of deviation from the linearity of the absorption [60].

#### Experimental Device for Column Tests

The experimental scheme of the apparatus used to perform the column tests is shown in Figure 1. The column is composed of a methacrylate tube with an inner diameter of 4.5 cm and a wall thickness of 0.5 cm, on the bottom it has a sheet of methacrylate that serves as a base. From this base a pipe for water flowing starts. This tube has a diameter of about 0.5 cm and is connected to a three-way valve that allows the treated water to exit from the column. On the outgoing water path, a small container of 50 mL was inserted to adjust the level of solution within the column. Moreover, in the lower part of the column we put a series of glass spheres, with a diameter of 1 mm so that the outlet water pipe has its inlet at a lower height than the overlying part of the column, where the reactive material was placed.

The main characteristic parameters of the column and of the reacting materials are summarized in Table 1.

The concentration of the solution used for the test was established taking into account the contamination threshold concentrations (CSC) in groundwater, set by the Italian legislation [61]. The transition metals were added in solution in the form of sulfated salts, respectively, of Zn, Cu and Cd.

The concentrations of these metals were determined by ICP-MS (inductively coupled plasma mass spectrometry—ELAN^®^6000 from Perkin Elmer, Waltham, MA-USA). Before starting each test, we verified that the density of the reactive media has a compaction degree compatible with the water flow within a real aquifer and that the velocity filtration assume a value adequate to guarantee a contact time between the contaminated solution and the reagent medium, such as to ensure a complete process of contaminant degradation. Each column was filled with the corresponding reactive material, with the density values reported in Table 1. Each column test was carried out using a mixture of zinc, copper and cadmium as main transition metals ion. Contaminated solution was set to flow, with the assigned concentration of the considered transition metals ion, from the top to the bottom of the column through a peristaltic pump connected by tygon tubes with the column and various containers (see Figure 1). During the test, samples were collected at fixed times at the exit of the column to obtain the variation trend of the concentration and evaluate the removal capacity of the reactive material over time. Initially, samples were collected twice a day; subsequently, the sampling frequency was reduced to once every two days and, at the end, to once a week. Furthermore, the pH of each sample was also measured as previously specified. Furthermore, to evaluate the adsorption capacity of each reactive material considered at the inlet and outlet of the column all reactive media were further characterized by field emission scanning electron microscopy (FESEM) (Huntington Beach-CA-USA) and energy dispersive X-ray analysis (EDX) to obtain high-resolution images of the surfaces, and therefore morphological information. For this purpose, the electron microscope model FESEM, SHIMADZU (NEON 40) was used.

### 3.6. Determination of the Permeable Reactive Barrier Thickness

One way to quantify the removal capacity of reactive medium can be determined by λ this parameter is known as the kinetic constant of degradation (m^3^/kg s), it allows relating the concentration of the pollutant in question, in this case heavy metals, with the rate of removal or better with the change of concentration over time. In other words, the velocity of metal removal, namely the variation of concentration over time, can be expressed by the following relation:(10)$$\frac{dC}{dt}=-\lambda \rho_{m}C$$
where *ρ_m_* the density (kg/m^3^), *t* the contact time (s) between the contaminant and the reagent and the meaning of *C* has already been defined before. Equation (10) is defined when *λ* is known. The variable *λ* value can be obtained, in the experimental phase, based on the values of the concentrations of the individual metals taken over time at the discharge of each column test.

By integrating Equation (10), the time *t* of residence (s) within the reactive barrier can be estimated as:(11)$$t=-\frac{1}{\lambda \rho_{m}}ln\left(\frac{C}{C_{0}}\right)$$

This time, *t* represents the contact time between the contaminated pore water and the reactive material that must be fixed to guarantee an adequate residence time of the contaminant within the PRB to be treated namely to reduce the concentration of transition metals ion from the initial value *C*_0_ to the target value clean up concentration *C*. Knowing the residence time, the porosity of the reagent and the flow rate passing through the barrier, it is possible to determine the PRB volume and, hence, knowing the transversal area crossed, it is possible to define the thickness *S* of the barrier according to the following relationship which is also commonly used to size the thickness of a PRB:(12)$$S=\frac{V}{A}=\frac{ki}{\lambda \rho_{b}}\;ln\left(\frac{C_{0}}{C}\right)$$
where *S* is the barrier thickness in the flow direction (m), *V* the volume (m^3^), *A* the transversal area (m^2^) of the barrier in the groundwater flow direction, *k* the hydraulic conductivity of the aquifer (m/s) (surrounding the barrier), *i* the piezometric gradient (--), *λ* the kinetic constant of degradation (m^3^/kg s), *ρ_b_* the bulk density (kg/m^3^), *C*_0_ the initial concentration (kg/m^3^), *C* the final concentration (kg/m^3^), namely the reclamation target. Equation (12) shows that the *λ* parameter can be obtained on the basis of the concentration values of the transition metals ions detected in samples collected at different times, during the column test. Therefore, in the experimental phase the *λ* constant parameter is obtained, by inverting the Equation (12), from the following equation:(13)$$\lambda =\frac{ki}{S\rho_{b}}\;ln\left(\frac{C_{0}}{C}\right)$$

Applying this relationship, we noted that the thickness of the barrier in the flow direction can be represented by the height of the column, assuming the bulk density equal to the density of the reactive material within the column, the hydraulic conductivity has the value obtained from the permeability tests performed. We set the gradient value to the one associated with the value of the obtained permeability from the column tests and associated with the value of the filtration rate. Regarding the concentration we employed the initial concentration *C*_0_ of the solution introduced into the column and the values obtained at the output of the column at the different times considered during the test.

### 3.7. Simulation of Mass Transport in Reactive Porous Media

The column tests carried out in laboratory were also simulated using the software RETRASO (reactive transport of solutes) [48]. This code solves the transport equations and the equations of chemical reactions simultaneously. The transition metals ion transport, based on mass balance of solute per unit of volume of saturated medium, is described by the following Equation (14) [62]:(14)$$\varepsilon \frac{\partial C}{\partial t}=G\frac{\partial C}{\partial z}-\varepsilon \cdot \alpha \cdot v\frac{\partial C}{\partial z^{2}}+R$$
where *C* is the total concentration in solution (kg/m^3^), *G* is the volumetric flow (m/s), *ε* is the effective porosity (-), *α* is the dispersivity (m), *v* is the velocity of the fluid within the reactive material (m/s), *R* is the *source term* (kg/m^3^s), which in this case is the balance between the two phases, known from the Langmuir equation, *t* is the time (s) and *z* indicates the value of the variable parameter on the vertical axis (namely the thickness of the reactive material within the column) (m) [49,63]. The chemical equations solve the aqueous speciation and the mass transfer of each component between the aqueous phase and the exchange complex. The input parameters required by the code RETRASO are the initial concentrations of the transition metals ion, the quantity of metal ion adsorbed at an assigned time, the maximum flow rate, the effective porosity, the height of the column and the simulation time extension. To physically base the simulations, we used data stemming from the results of our column tests. The simulation code provides the variation of transition metals ions considered over time and, consequently, the times of maximum adsorption and saturation of the solution. This software is able to provide the breakthrough curve and, therefore, to identify the breaking point (break point), namely the time when the effluent concentration exceeds the objective concentration, and also the point of exhaustion (exhaustion point), which represents, the time when the initial concentration of the effluent is equal to the final concentration and the reactive material is no longer capable to eliminate the transition metals ion. Our model does not consider reactive transfer solute.

## 4. Results and Discussion

The hydraulic conductivities, *k*, were determined in laboratory only for the materials which provided larger adsorption kinetics, namely cabuya fibers and natural zeolites. We estimated *k* values by setting three different hydraulic head values. As a result, we obtained for the cabuya fibers an average value of *k* = 1.33 × 10^−3^ m/s.

This value is lower than that obtained by Mayacela–Rojas et al. [46] for cabuya fiber (*k* = 4.95 × 10^−4^ m/s). This is because the hydraulic conductivity values were strongly influenced by the degree of compaction of the material put into the cell. Specifically, with reference to vegetable reactive materials, previous studies [25,29] showed that hydraulic conductivity is inversely proportional to the density. The value of *k* for the three natural zeolites we considered has an average value of 1.18 × 10^−3^ m/s, with an order of magnitude in agreement with previous studies for zeolites of similar characteristics. For the proper design of a PRB it is important to use a reactive material with a hydraulic conductivity larger than the surrounding aquifer [45,46].

For this reason, the granular materials are very suitable for real applications. The value of the density of the cabuya fibers used for the *k* measures was set to 16 kg/m^3^ [45,46]. The values of total and effective porosity, estimated for the two types of mediums considered, are shown in Table 2.

Table 2 shows that with the fiber density assumed, the porosity of the cabuya is larger than that of the zeolites considered. The porosity values of the various zeolites employed were significantly coincident. Moreover, for the reactive materials considered the following pH values were measured: 7–8.15 for cabuya fiber, 8–8.01 for clinoptelolite natural zeolite, 7–8.09 for mordenite natural zeolite and 9–9.41 for sodium mordenite natural zeolite. These values show that both cabuya fibers and mordenite natural zeolite have an almost neutral pH, while clinoptilolite natural zeolite and sodium mordenite natural zeolite have a basic character. In acidic environments the functional groups act positively, which causes less attraction between metals and minerals, whereas in basic environments the functional groups behave as negatively charged elements, attracting heavy metals. The lowest adsorption occurs in an acidic environment (pH ≤ 3), on the other hand, the highest adsorption commonly occurs in a pH range of 5.0–8. The most promising PRBs are those in which pH changes are not significant [7]. Metal sorption processes: equilibrium and kinetic characterization with batch tests.

Batch tests were performed for all of the seven reactive media considered to identify the main adsorption features of the substances investigated and define which reactive mediums are more suitable for a PRB and hence for column tests. Obtained results highlighted analogous behaviors for the different materials with similar decreasing trends associated with different performances in terms of quantity and velocity of removal. Figure 2 shows the results of batch tests with observed concentrations trends over time, for each reactive material and for all transition metals considered with percentages of absorption of 90%. Results stemming from our tests are in line with those obtained by Molinari et al. [47], where high adsorption percentages (>90%) of Cu, Cd and Zn were observed during the execution of the Batch tests, using cabuya fiber and natural zeolites as reactive media. Additionally, important removal yields of these heavy metals were identified after approximately 20 h, a time very similar to that obtained in this research.

In fact, cabuya fiber and natural zeolites exhibited similar concentration trends, while calcareous limestone, synthetic zeolite 4A and ZVI observed different trends which highlights the need of larger contact time to observe significant abatements of dissolved transition metals concentration. In detail, all the reactive materials considered showed similar adsorption kinetics, with removal capacity close or larger than 90%. The best performances were exhibited by natural zeolites (95% of transition metals ions removed in about 10 h). In other words, the zeolite that shows the highest ion exchange capacity is the one that exhibits the highest percentage of removal for that ion [64,65,66,67,68,69]. Specifically, against the Cu, natural zeolites (especially mordenite) and cabuya fibers presented the largest removal percentages, while the calcareous limestone showed lower adsorption percentages even if still high. Synthetic zeolite and ZVI showed the lowest values.

Regarding Cd, removal percentages above 90% were observed for all the materials investigated. Moreover, for Zn the removal percentages were very close to 100% for all the reactive materials mediums employed for our tests, except for the calcareous limestone, which, however, exhibited an adsorption percentage of 93%.

Table 3 shows the amount of metal adsorbed per unit of adsorbent mass in equilibrium conditions (*q_max_*) estimated by means of the adsorption isotherms for each reactive material and for each transition metals ion considered [66]. The data in Table 3 show that the quantity of adsorbed metal ions (*q_max_*) is different for the three transition metals considered. In particular, for all the reactive materials examined, the *q_max_* value is minimal for Cd (5.2–5.3 mg/g), while for Cu it is higher, albeit not much (5.5–6.9 mg/g). The higher value of *q_max_* was obtained for the Zn (11.3–13.3 mg/g), for which the parameter under examination was found to be about double compared to Cu and more than double compared to Cd. These results highlighted that the most reactive materials, with a high *q_max_* value, are, respectively, cabuya fiber, natural clinoptilolite zeolite, natural mordenite zeolite and sodium mordenite natural zeolite. Considering that calcareous limestone, ZVI and synthetic zeolite 4A exhibited longer adsorption kinetics, hence larger contact times to begin the adsorption of the considered transition metals, with respect to the other reactive media employed within the present study, we selected, for column tests, only cabuya fibers and the three types of natural zeolites, namely clinoptilolite, mordenite and sodium mordenite disregarding calcareous limestone, synthetic zeolite 4A and ZVI [47].

### 4.1. Determination of Adsorption Isotherms

For the four reactive materials selected, the phenomena related to the adsorption capacity of the solid phase were deepened.

The equilibrium conditions of the pollutant adsorbed in solution and in solid phase were defined by the equilibrium isotherms of Langmuir and Freundlich described, respectively, by the Equations (8) and (9) [59,60]. Table 4 shows the characteristic parameters of the equilibrium isotherms considered for each reactive material and for the transition metals examined.

Where the meaning of $q_{m}$, *K_L_*, *K_f_* and n was previously defined and R^2^ is the determination coefficient. The R^2^ values are all high and close to unity, so the models taken into consideration, represented by the equations (8) and (9), can be considered significantly representative of the trend of $q_{e}$ vs $C_{e}$. For the Langmuir isotherm, the equilibrium parameter RL predicts the affinity between the adsorbate and the adsorbent. The Langmuir isotherm is irreversible (RL = 0), favorable (RL < 1), linear (RL = 1) or unfavorable (RL > 1) [59]. Our analysis found that for all the selected reactive media RL was less than 1, and therefore, the adsorption system is favorable. According to Mishra et al. [67], the R^2^ value allows to know the goodness of the model based on the analysis of the adsorption variables. In the case of Langmuir isotherm, the analyzed variables are (1Ce) (x) and (1qe) (y), and for the Freundlich isotherm, those used are $LnC_{e}$(x) and $Ln\;q_{e}$ (y). In our study the R^2^ values for all cases is close to 1, which evidences a strong correlation between variables. In all cases considered, the values of the Freundlich exponent (n), were larger than 1 and less than 10, indicating that the heavy metals were favorably adsorbed by each reagent substance used [60]. On the other hand, the values of the root mean square error (RMSE) calculated for each of the reactive media has always a value close to zero, which supports the validity of the models proposed, both for the Langmuir isotherm and the Freundlich isotherm.

### 4.2. Metal Sorption Processes: Equilibrum and Kinetic Characterization with Columns Tests

On the basis of batch tests outcomes, we selected for materials for column evaluation: cabuya fiber, clinoptilolite natural zeolite, mordenite natural zeolite and sodium mordenite natural zeolite. The column tests, performed by the laboratory apparatus depicted in Figure 1 and conducted in continuous for durations spanning from about five days to about two months and a half, allowed to determine the process parameters under dynamic flow conditions and therefore in conditions closer to the real applications. With the column test we determined the removal efficiency of the transition metals ions considered (Zn, Cu and Cd) for each reactive materials employed.

Table 5 reports the set of main parameters for column tests for the three transition metals tested together with the duration of each test.

Form the tests we observed that to achieve the saturation of the reactive materials investigated for all the metals considered, each test should have a duration between 133.61 h and 1793.75 h, namely between 5.57 days and 74.74 days, depending on the reactive material and transition metal considered.

It is important to note that the durations reported in Table 5 represent reference times and not definitive times since the *q_max_* used was obtained by batch tests performed under static conditions, while the column tests were performed considering a continuous flowing. The column tests were continued for about 1200–2000 h (about 50–83.3 days). At fixed times, samples were collected and subjected to: pH measurement, filtration, dilution and finally analysis by ICP-MS for the determination of the final concentration at the saturation time of each reactive material.

Figure 3, Figure 4 and Figure 5, respectively, depict the results of the column tests for Zn, Cu and Cd and for the four reactive materials considered. The reported results mark a trend and show that there is a higher adsorption of metals, in contact with the different reactive media, in the early contact times. The adsorption decreases with increasing time, without a significant change in the amount of adsorbed metal ions. This plateau represents saturation of the active sites available on the reactive medium samples for interaction with metal ions [26].

The saturation times obtained for each reactive material considered are shown in the following Table 6.

The maximum adsorption time indicate, for each reagent material employed, the time, in hours, beyond which the adsorption of the transition metals ion dissolved in solution begins to decrease. Total saturation time indicate the time, in hours, when the reactive material does not adsorb more contaminant and hence their reactivity can be considered finished.

Considering that the column tests had a final duration of about 2000 h, from Table 6 it is possible to identify the reactive materials that showed larger adsorption capacity in relation to the relative saturation time. These are, in descending order: mordenite natural zeolite, clinoptilolite natural zeolite, sodium mordenite natural zeolite and cabuya fiber.

#### 4.2.1. Characterization of the Metal Sorption Processes by FESEM and EDX Analysis

The results of the FESEM and EDX analyzes made, to evaluate the adsorption capacity of reactive materials considered, at the inlet and outlet of the column, are shown in Figure 6, Figure 7, Figure 8 and Figure 9a,b, respectively, related to each reactive material. The FESEM analysis of the cabuya fiber (Figure 6) allowed to determine the size of the fiber bundles, with an indicative value of 232.9 μm, also highlighting a typical internal morphology with empty spaces.

The brightness of some solids confirms the presence of the investigated transition metals. This is also confirmed by the EDX spectrum, identifying the presence of Zn (with a percentage of 20.5%) and Cu (with a percentage of 39%). The graphs of Figure 7, Figure 8 and Figure 9, relating to the natural zeolites, show the presence of Zn and Cu in the relative spectra. Specifically, the percentages of Zn and Cu obtained were, respectively: for the Clinoptilolite natural zeolite 15% and 23%; for the mordenite natural zeolite 23% and 39%; for sodium mordenite natural zeolite 13% and 19%. Cd was not detected because, probably, the initial concentration was too low to observe its presence on the reactive media. Examining these results, we observed that the mordenite natural zeolite retained a larger number of metallic cations than the other reactive materials considered, in agreement with the results obtained from the batch tests which show for Zn a *q_max_* of 13.3 mg/g and for Cu a *q_max_* of 6.9 mg/g.

#### 4.2.2. Estimation of PRB Thickness Using Tested Reactive Materials

The calculation of the thickness of the PRB was made, for all the selected reactive materials and for all the transition metals considered (Zn, Cu and Cd), using Equation (12), where all parameters are known as reported in Table 7, together with the values obtained for the thickness of the PRB. From Table 7 we observe that to achieve target concentrations, the PRB should have a thickness of at least 1.1 m for the cabuya fiber, 2.1 m for the clinoptilolite natural zeolite, 1.01 m for the mordenite natural zeolite and 1.9 m for sodium mordenite natural zeolite.

#### 4.2.3. Simulation Results by RETRASO Code

The simulation by the RETRASO code was made for all the selected reactive materials and considering all the transition metals taken into consideration (Zn, Cu and Cd), using Equation (14). Groundwater flow and contaminant transport models are essential for the study of the processes occurring in the water-soil matrix. They also allow to predict the behavior of the PRB, having as input data the laboratory tests performed [48]. For this purpose, for all the reactive materials and transition metals ions considered, the parameters shown in Table 8 were used.

On the basis of the results obtained for the reactive materials investigated with the column tests, their behavior was extrapolated for a PRB with a thickness of 1 m and unitary section. Therefore, it was possible to determine the necessary time to achieve the required target concentration (legal limit) for each of the polluting materials considered and the maximum useful operating time of the PRB [48]. Table 9 shows the maximum adsorption times for each reactive material and the time, in days, beyond which the adsorption of the transition metals ion considered starts to decrease.

It is appropriate to specify that the above results were obtained for assigned values of the flow rate and initial concentrations of the three transition metals ions considered.

To take into account values of flow rate, initial concentrations and/or barrier thickness, different from those considered here, it is necessary to repeat the calculation.

Obviously, for low values of transition metals ion concentrations the saturation times will be greater.

Similarly, for higher flows, saturation times will be lower and, moreover, for PRB thicknesses greater than the one considered above, higher saturation times will be obtained.

The values of Table 9 show that the maximum adsorption times of the cabuya fiber are much lower than those obtained for the other zeolitic materials examined.

Similar to zinc, copper and cadmium, the cabuya fiber has saturation times of 125, 63 and 41 days, respectively, while for the other reactive materials examined, namely the zeolites, this time varies between 900 days and a duration of more than 1300 days, that is between about 2.5 years and a time certainly greater than 4 years, depending on the transition metals ion considered.

This is an indication of an adsorption capacity of the cabuya fiber evidently lower than the other reactive materials examined.

From the results obtained it can be noticed that among the materials examined the most reactive one for the realization of a PRB is the mordenite natural zeolite, which has a higher saturation time, and therefore an adsorption capacity, higher than the others. However, when constructing an in situ PRB with natural zeolites, it should be considered that the removal capacity of the zeolite can be influenced by the pH of the surrounding medium and its components, such as Ca, Mg, Na, SO_4_ 2, CO_3_ 2 and dissolved organic matter. Zeolite efficiency can also be affected by groundwater constituents that compete with contaminants for binding sites [16].

Figure 10a–d show the adsorption trend and the saturation achievement, respectively, for the cabuya fiber, the clinoptilolite natural zeolite, the mordenite and the sodium mordenite.

Regarding the Cd, it should be specified that from these graphs it is not possible to identify the adsorption trend and the saturation time, since the values of the initial concentration for this material are lower than those used for zinc and copper. Therefore, for the Cd the corresponding values of maximum adsorption and saturation times are certainly higher than those obtained for Zn and Cu.

## 5. Future Challenges

PRBs have proven to be a good alternative for the treatment of contaminated aquifers, when compared to the traditional pump and treat technique. However, on several occasions this technique is not the most chosen, An et al., 2016 [70] demonstrates this fact by comparing the four most known remediation techniques and ranks them in order of importance considering the following variables: social, economic and technological aspects, and the following sequence of sustainability was obtained: natural attenuation > pumping and treatment > PRB > air sparging [71]. Therefore, future research could be aimed at making the use of PRBs more sustainable by emphasizing aspects such as: precipitation of reactive materials and increasing the duration of PRBs at the time of installation.

On the other hand, good results have been obtained by using, in addition to PRBs, the coupled use of two or more remediation techniques (use of microorganisms, electrokinetic adsorption barrier, soil washing, among others), therefore, further research in this area may be of collective benefit to make the implementation of this technique more promising [72,73,74].

## 6. Conclusions

With the focus on remediation of contaminated sites, regulatory agencies are increasingly turning to the application of new, less costly and more effective technologies. When selecting cleanup alternatives, a significant benefit-cost ratio can be demonstrated for PRB over a P&T system (or any other competitive technology) [75]. Case studies of full-scale PRBs are very limited and mostly concentrated in developing countries, however, this remediation technique could become a good option, especially for developing countries, due to its low investment for operation and maintenance [7]. Research aimed at analyzing and selecting new local reactive materials would contribute to a better management of available water resources.

In this context, in this study the reactive behavior of different materials against three transition metals was investigated to verify their suitability for the realization of a PRB for a field scale application. As reactive materials we considered natural fibers (i.e., cabuya fibers) and three natural zeolites (i.e., clinoptilolite, mordenite and sodium mordenite), while the transition metals ions employed were a solution of zinc, copper and cadmium.

All reactive media were deeply characterized to investigate the internal and superficial structure of each material and evaluate the mechanisms responsible of their reactivity. Reactive materials were characterized by batch tests and subsequently by column tests, which were also modeled by RETRASO code. From our tests we obtained the following major findings:

Batch tests confirmed the significant adsorption capacity of the tested reagents against all transition metals considered.

From the analysis of the adsorption kinetics, we observed that the use of a second order adsorption kinetic provides a more reliable description of the experimental behaviors observed at laboratory scale with specific regards to the reaction velocity of the various materials than the use of a first order kinetic. The use of Langmuir and Freundlich adsorption isotherms showed, respectively, the presence of affinity conditions between adsorbed and adsorbent and a favorable adsorption for each reactive material. Column tests allowed to determine experimentally, in situations close to the real ones and for each reactive material considered, the maximum adsorption and saturation times. The largest adsorption time (i.e., 833 h) was obtained in the case of mordenite natural zeolite, while the minimum one (i.e., 189 h) was observed in the case of cabuya fiber. Mordenite natural zeolite exhibited the maximum saturation time (i.e., 1826 h), while cabuya fiber provided the minimum value (i.e., 515 h). These results highlighted that, among the four reactive materials considered, the most suitable and promising in term of contaminant sequestration by means of a PRB is the mordenite natural zeolite.

On the basis of the experimental results, we evaluated the possible thicknesses of a real PRB. The thickness values obtained for mordenite, in case of all transition metals considered, was the lowest with respect to the other reactive materials considered. This finding provides further confirmation that mordenite zeolite is the most reactive material exhibiting the largest adsorption capacity towards the transition metals investigated.

A verification of the PRB sizing was also carried out by using the RETRASO code, simulating the column tests for all the four reactive materials considered and assuming 1 m column length, which is an intermediate value among those obtained from experimental results. The RETRASO simulations confirmed the lower reactivity of the cabuya fibers with respect to the zeolitic materials considered. Among the zeolitic materials explored, the largest adsorption capacity, against all the three transition metals examined, was exhibited by the mordenite natural zeolite.

Amongst the four media investigated in the present study the mordenite natural zeolite seems to be the most suitable for the realization of a PRB at field scale instead of the classical zero valent iron commonly employed. On the other hand, the lower saturation time exhibited by the cabuya fiber against the zeolites can represent a stimulus for the development of new methodologies that do not require high saturation times, also considering that natural fibers can provide important advantages such as the ease of retrieval, very low cost of supply, easiness of density regulation within the PRB.

Despite our study shed lights on the possible use of different reactive media for a PRB than the classical medium, many other aspects remain to be investigated to improve the comprehension of the behavior of these media and generalize our results. Hence, future investigations should be, for example, devoted to the comprehension of the influence exerted by the density of the reagent, which in turn influence porosity and hydraulic conductivity, by focusing the attention to the comprehension of the role of the interconnection between the reactive material filling the PRB and the site-specific characteristics of the porous medium housing the aquifer that need to be reclaimed.

## Figures and Tables

**Figure 1 ijerph-18-06075-f001:**
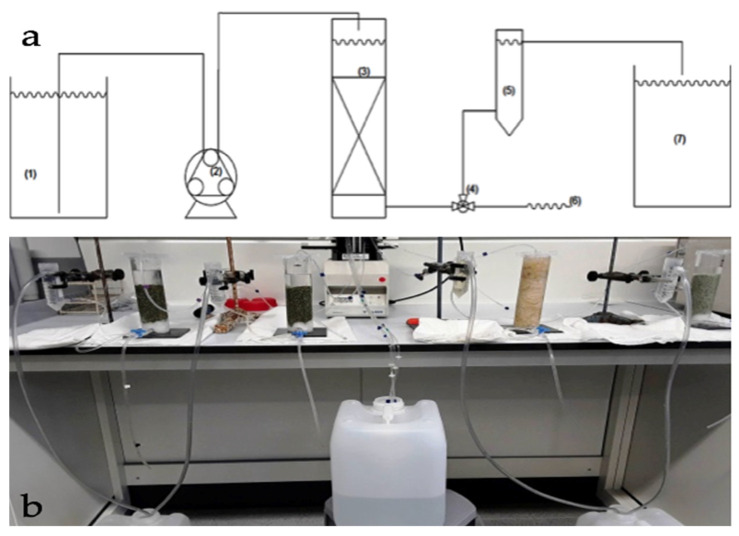
(**a**) Outline of the system used to perform column tests. (1) Reservoir; (2) peristaltic pump; (3) column; (4) three-way valve; (5) container controller of the liquid level in the column; (6) sampling; (7) collector output from the column. (**b**) Photo of the laboratory apparatus used for column tests.

**Figure 2 ijerph-18-06075-f002:**
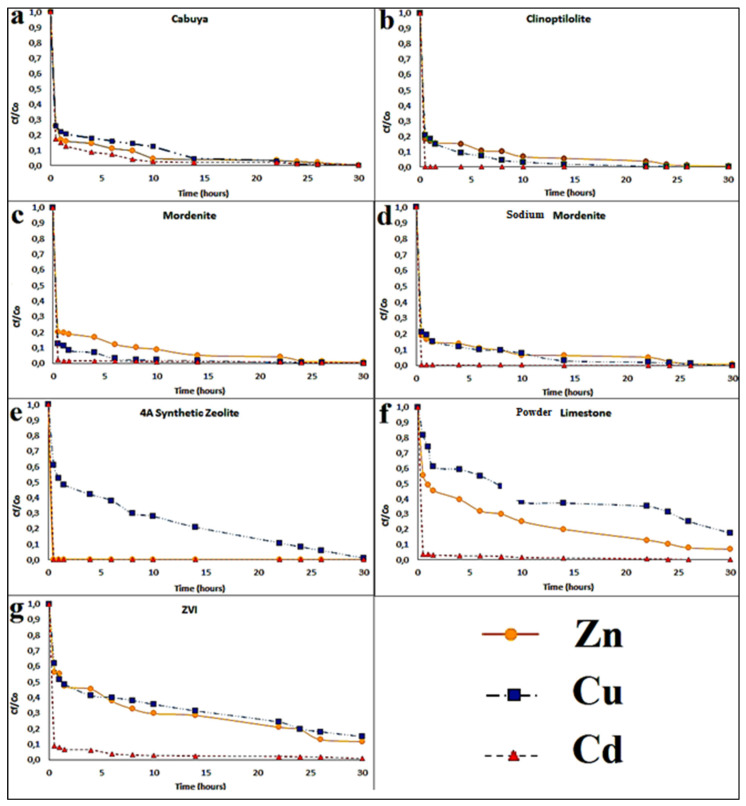
Concentration trends over time, for each reactive material and for all transition metals considered, resulting from the batch tests. The experimental conditions in ranges were: static fluid, temperature: 25–28 °C and pH: 7.1–8.8. Specifically: (**a**) for Cabuya; (**b**) for Clinoptilolite; (**c**) for Mordenite; (**d**) for Sodium Mordenite; (**e**) for 4A Sinthetic Zeolite; (**f**) Powder Limestone; (**g**) Zero Valent Iron (ZVI).

**Figure 3 ijerph-18-06075-f003:**
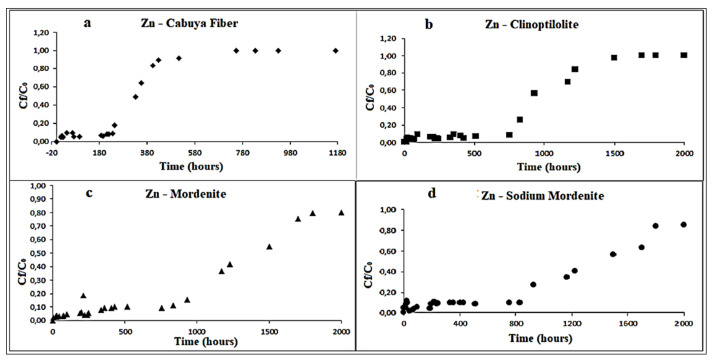
Variation of Zn concentration over time, in the column tests, for the following reactive materials: (**a**) cabuya fiber; (**b**) clinoptilolite zeolite; (**c**) mordenite zeolite; (**d**) sodium mordenite zeolite. The experimental conditions in ranges were: dynamic fluid, temperature: 26–28 °C and pH: 7–8.3.

**Figure 4 ijerph-18-06075-f004:**
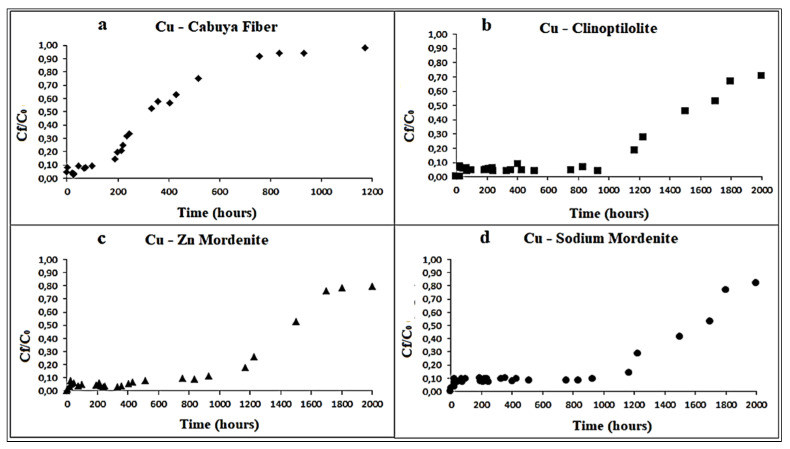
Variation of Cu concentration over time, in the column tests, for the following reactive materials: (**a**) cabuya fiber; (**b**) clinoptilolite zeolite; (**c**) mordenite zeolite; (**d**) sodium mordenite zeolite. The experimental conditions in ranges were: dynamic fluid, temperature: 25–27 °C and pH: 7–8.6.

**Figure 5 ijerph-18-06075-f005:**
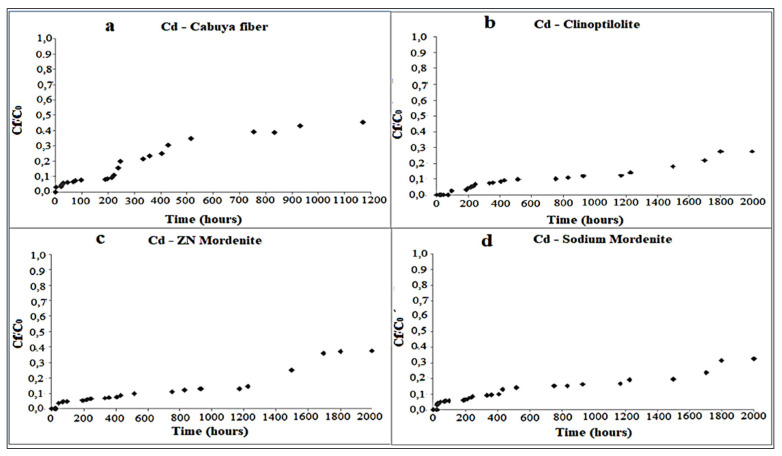
Variation of Cd concentration over time, in the column tests, for the following reactive materials: (**a**) cabuya fiber; (**b**) clinoptilolite zeolite; (**c**) mordenite zeolite; (**d**) sodium mordenite zeolite. The experimental conditions in ranges were: dynamic fluid, temperature: 26–28 °C and pH: 7.4–8.8.

**Figure 6 ijerph-18-06075-f006:**
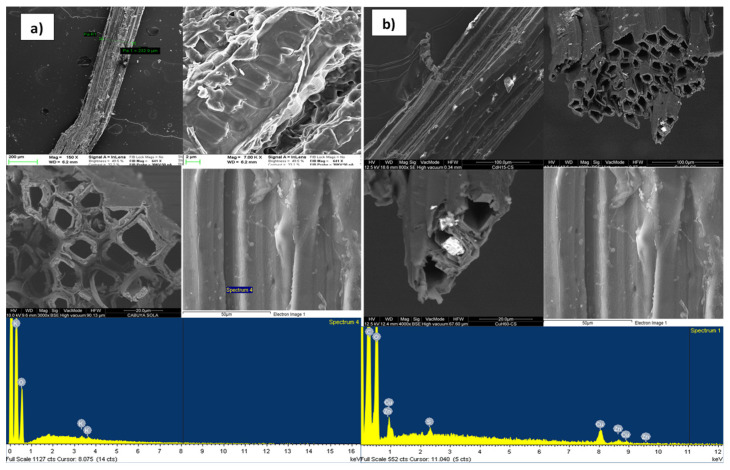
FESEM and EDX microphotographs of the cabuya fiber, (**a**) inlet of the column. (**b**) outlet of the column.

**Figure 7 ijerph-18-06075-f007:**
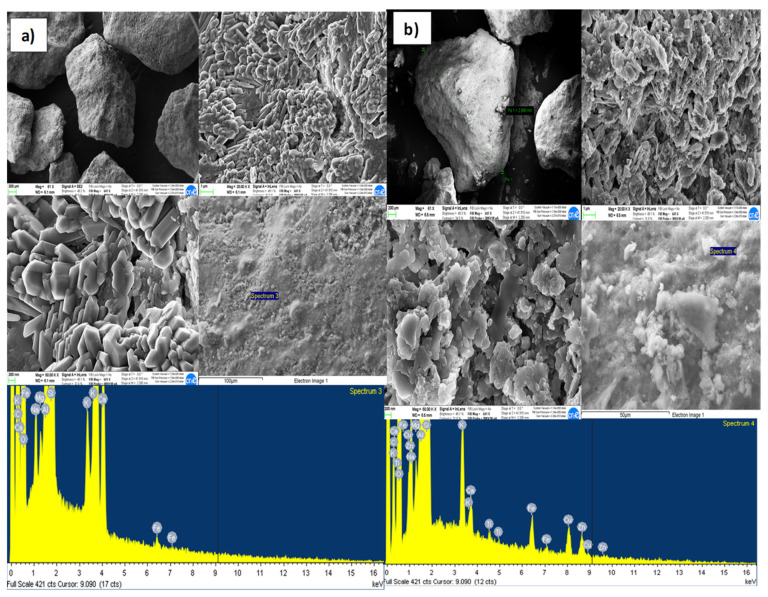
FESEM and EDX microphotographs of the clinoptilolite natural zeolite (**a**) Inlet of the column. (**b**) Outlet of the column.

**Figure 8 ijerph-18-06075-f008:**
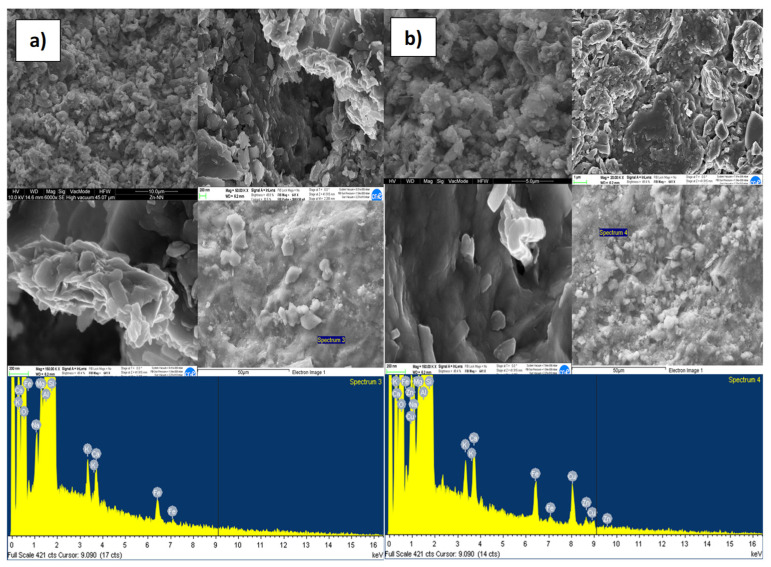
FESEM and EDX microphotographs of the mordenite natural zeolite, (**a**) inlet of the column. (**b**) Outlet of the column.

**Figure 9 ijerph-18-06075-f009:**
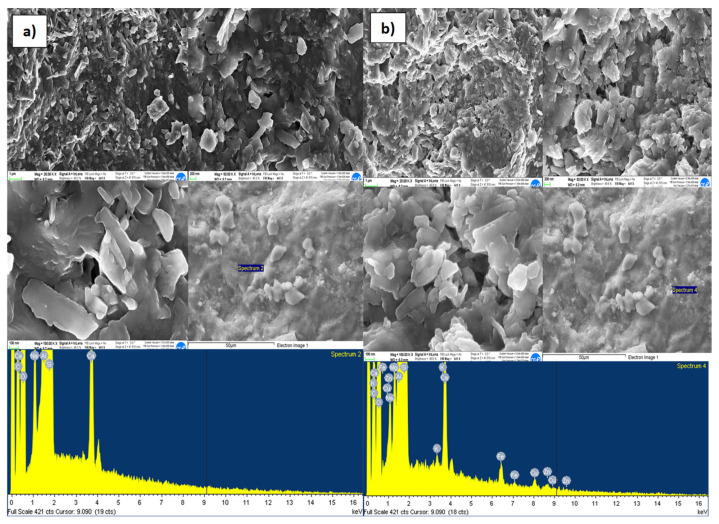
FESEM and EDX microphotographs of the sodium mordenite natural zeolite, (**a**) inlet of the column. (**b**) Outlet of the column.

**Figure 10 ijerph-18-06075-f010:**
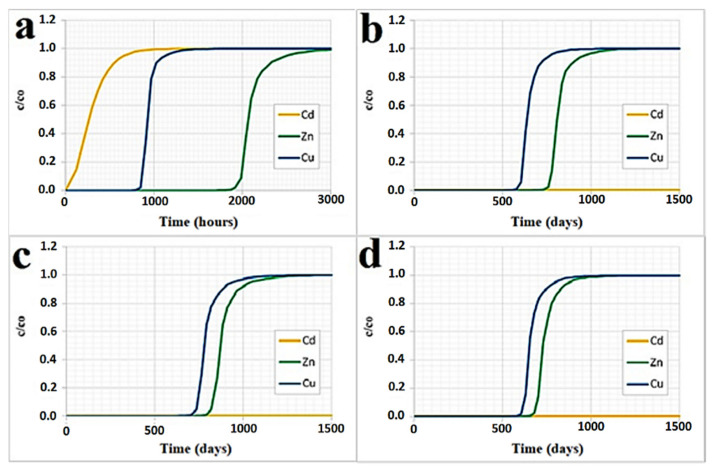
Adsorption trend and saturation times, simulated by the code RETRASO for all transition metals considered, respectively, in the case of: (**a**) cabuya fiber; (**b**) clinoptilolite natural zeolite; (**c**) mordenite natural zeolite; (**d**) sodium mordenite natural zeolite.

**Table 1 ijerph-18-06075-t001:** Experimental operation details using column tests.

Parameters	Unit of Measure	Cabuya Fiber	Zeolites
			Clinoptilolite Mordenite Sodium Mordenite
Effective porosity		0.55	0.30
Density	kg/m^3^	16	100
Mass of the reactive material	kg	0.030	0.083
Column diameter	m	0.042	
Column area	m^2^	0.0014	
Height of the reactive column	m	0.18	
Reactive column volume	L	25	
Metal concentration	mg/L	Zn: 100; Cu: 30; Cd: 4	
Flow rate	L/h	0.0216	
Velocity	m/h	0.016	

**Table 2 ijerph-18-06075-t002:** Total and effective porosity values of the reactive materials considered.

Reactive Material	Total Porosity	Effective Porosity	
		Densimetric Method	Tracking Method
Cabuya fiber	0.84	0.55	0.56
Natural zeolite	0.62	0.30	0.30

**Table 3 ijerph-18-06075-t003:** Amount of metal adsorbed per unit of adsorbent mass under equilibrium conditions.

Reactive Material	Zn			Cu			Cd		
	Adsorption %	Contact Time (*h*)	*q_max_* (mg/g) (*C_0_* = 100 mg/L)	Adsorption %	Contact Time (*h*)	*q_max_* (mg/g) (*C_0_* = 50 mg/L)	Adsorption %	Contact Time (*h*)	*q_max_* (mg/g) (*C_0_* = 40 mg/L)
Cabuya fiber	100	30	13.3	99	24	6.7	99	24	5.2
Clinoptilolite zeolite	99	26	13.3	99	22	6.7	99	26	5.3
Mordenite zeolite	99	24	13.2	99	22	6.9	99	8	5.3
Sodium mordenite zeolite	99	30	13.2	99	26	6.7	99	22	5.3
Calcareous limestone	93	30	12.4	82	30	5.5	100	24	5.3
4A Synthetic zeolite	85	30	11.3	99	30	6.5	100	26	5.2
ZVI	88	30	11.8	85	30	5.7	99	30	5.3

**Table 4 ijerph-18-06075-t004:** Characteristic parameters of Langmuir and Freundlich isotherms.

Metal	Reactive Materials	Langmuir					Freundlich			
		*q_m_*	*K_L_*	*RL*	R^2^	RMSE	*K_f_*	*n*	R^2^	RMSE
		(mg/g)	(mg/L)				(L/g)	(g/L)		
**Zn**	Cabuya fiber	11.20	0.74	0.016	0.97	0.010	5.33	4.21	0.95	0.071
	Clinoptilolite	12.53	3.13	0.004	0.95	0.010	8.00	5.12	0.88	0.020
	Mordenite	14.00	2.10	0.006	0.88	0.014	9.10	6.58	0.92	0.072
	Sodium mordenite	13.40	0.63	0.018	0.90	0.009	8.00	8.23	0.80	0.155
**Cu**	Cabuya fiber	16.47	0.11	0.123	0.99	0.015	3.42	2.29	0.95	0.025
	Clinoptilolite	15.60	0.59	0.028	0.96	0.029	4.21	2.41	0.93	0.188
	Mordenite	17.17	0.20	0.074	0.99	0.008	2.80	1.50	0.97	0.888
	Sodium mordenite	12.74	2.96	0.006	0.94	0.046	6.22	3.98	0.78	0.382
**Cd**	Cabuya fiber	7.67	0.41	0.152	0.99	0.172	1.66	1.83	0.98	0.192
	Clinoptilolite	15.58	0.13	0.287	0.99	0.019	2.65	2.29	0.92	0.364
	Mordenite	6.10	0.46	0.145	0.91	0.090	1.41	1.57	0.99	0.121
	Sodium mordenite	18.52	0.05	0.440	0.99	0.020	1.11	1.40	0.99	0.118

**Table 5 ijerph-18-06075-t005:** Main parameters for sorption column tests for Zn(II), Cu(II) and Cd(II) transition.

Metals	Reactive Material		Area (m^2^)	Mass (g)	Concentration (mg/L)	*q_max_* (mg/g)	Mass Held by the Column (mg)	Dissolution Volume (L)	*Q* (L/h)	Test Duration Days Hours		Velocity (m/h)
Zn	Fiber	Cabuya	0.0014	30	100	9.62	288.60	2.89	0.022	5.5	133.6	0.016
	Zeolites	Clinoptilolite		83		10.40	863.20	8.63		16.65	399.63	
		mordenite		83		10.42	864.86	8.65		16.68	400.00	
		Sodium mordenite		83		9.89	820.87	8.21		15.83	380.03	
Cu	Fiber	Cabuya	0.0014	30	30	3.10	93	3.10	0.022	5.98	143.52	0.016
	Zeolites	Clinoptilolite		83		3.25	269.75	8.99		17.35	416.28	
		mordenite		83		3.15	261.45	8.72		16.81	403.47	
		Sodium mordenite		83		3.28	272.24	9.07		17.51	420.12	
Cd	Fiber	Cabuya	0.0014	30	4	1.09	32.7	8.18	0.0216	15.77	378.47	0.016
	Zeolites	Clinoptilolite		83		1.79	146.78	36.70		70.78	1698.84	
		Mordenite		83		1.89	154.98	38.75		74.74	1793.75	
		Sodium mordenite		83		1.75	143.50	35.88		69.203	1660.88	

**Table 6 ijerph-18-06075-t006:** Maximum adsorption times and saturation times of Zn, Cu and Cd, for each reactive material considered.

Metals	Reactive Materials	Maximum Adsorption Times (Hours)	Saturation Times (Hours)
Zn	Cabuya fiber	237	755
	Clinoptilolite	357	1170
	Mordenite	429	1800
	Sodium mordenite	405	1700
Cu	Cabuya fiber	189	755
	Clinoptilolite	833	1820
	Mordenite	515	1826
	Sodium mordenite	429	1705
Cd	Cabuya fiber	245	515
	Clinoptilolite	405	1789
	Mordenite	833	1802
	Sodium mordenite	930	1718

**Table 7 ijerph-18-06075-t007:** Summary table of the hydraulic conductivity (*k*) values, piezometric gradient (*i*), first-order (*λ*_1_) degradation kinetics constant, bulk density (*ρ*), initial concentration (*C*_0_), final concentration (*C_f_*) which is the objective of the reclamation and thickness (*S*) of the PRB.

Reactive Material	Metals	*K* (m/day)	*i*	*λ*_1_ (L/g h)	*λ*_1_ (L/g day)	*ρ* (kg/m^3^)	*C*_0_ (kg/m^3^)	*C*_f_ Limit (kg/m^3^)	ln(*C*_0_/*C*_f_) (--)	*S* (m)
Cabuya fiber	Zn	141.4	0.913	0.105	2.517	16	0.100	0.003	3.507	1.1
	Cu			0.152	3.651		0.030	0.001	3.401	0.8
	Cd			0.149	3.576		0.004	0.00005	4.382	1.0
Clinoptilolite	Zn	131.6	0.913	0.124	2.971	100	0.100	0.003	3.507	1.0
	Cu			0.190	4.559		0.030	0.001	3.401	0.6
	Cd			0.072	1.726		0.004	0.00005	4.382	2.1
Mordenite	Zn	131.5	0.913	0.145	3.479	100	0.100	0.003	3.507	0.8
	Cu			0.185	4.445		0.030	0.001	3.401	0.6
	Cd			0.140	3.351		0.004	0.00005	4.382	1.1
Sodium mordenite	Zn	131.5	0.913	0.137	3.278	100	0.100	0.003	3.507	0.9
	Cu			0.109	2.625		0.030	0.001	3.401	1.1
	Cd			0.080	1.917		0.004	0.00005	4.382	1.9

**Table 8 ijerph-18-06075-t008:** Input parameters of simulation by RETRASO code.

Parameters	Metals	Cabuya Fiber	Clinoptilolite Natural Zeolite	Mordenite Natural Zeolite	Sodium Mordenite Natural Zeolite
*C**_input_* (mg/L)	Zn	100	100	100	100
	Cu	100	100	100	100
	Cd	1	1	1	1
*q_max_* (mg/g)	Zn	18.3	22.9	33.8	26.7
	Cu	6.5	10.3	10.9	9.1
	Cd	1.1	1.7	1.7	1.5
Flow rate (L/min)		0.17	0.17	0.17	0.17
Effective porosity		0.55	0.30	0.55	0.30
Column height (m)		0.14	0.06	0.14	0.06
Simulation height (m)		1	1	1	1

**Table 9 ijerph-18-06075-t009:** Maximum adsorption and saturation times obtained by simulation with the RETRASO code for each reactive material and for the transition metals considered.

Metals	Reactive Materials	Maximum Adsorption Times (Days)	Saturation Times (Days)
Zn	Cabuya fiber	83	125
	Clinoptilolite	800	1200
	Mordenite	429	1300
	Sodium mordenite	750	1200
Cu	Cabuya fiber	33	63
	Clinoptilolite	600	900
	Mordenite	750	1200
	Sodium mordenite	600	1000
Cd	Cabuya fiber	2	41
	Clinoptilolite	(>800)	(>1200)
	Mordenite	(>750)	(>1300)
	Sodium mordenite	(>750)	(>1200)

## Data Availability

The data used in this paper were generated during the study. For any other data, please refer to the works indicated in the References.

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
