# Peer review of "Removal of Transition Metals from Contaminated Aquifers by PRB Technology: Performance Comparison among Reactive Materials"

_ijerph, 2021, doi:10.3390/ijerph18116075_

Round 1

Reviewer 1 Report

Dear Editor,

Regarding my suggestions, all of them have been properly addressed.

Some small (important) points:

L 313 Unit is spelled wrong.

L 339 Unit is spelled wrong. Check it in whole manuscript.

Figures still demands improvements.

Figure 4: Attemption! C/Co is dimensionless, not in mg/L.

Sincerely yours.

Author Response

Reviewer 1

Suggestions

Change

Line

L 313 Unit is spelled wrong.

L 339 Unit is spelled wrong. Check it in whole Figure 4: Attemption! C/Co is dimensionless, not in mg/L.manuscript.

All SI units have been reviewed and changes have been made.

Figures still demands improvements.

Graphics were improved

Reviewer 2 Report

The authors have adequately answered most of the questions raised by the reviewer. It only remains to be clarified why they consider degradation and degradation kinetic constant (equation 10), when there is not justified that a chemical reaction takes place in the barrier. Is there adsorption or reaction in the materials used in PBR?

Editing questions: Some Figure titles  are displaced (Figure 5)

Author Response

Reviewer  2

It only remains to be clarified why they consider degradation and degradation kinetic constant (equation 10), when there is not justified that a chemical reaction takes place in the barrier. Is there adsorption or reaction in the materials used in PBR?

The paragraph has been rewritten using only the term degradation kinetic constant to avoid confusion.

L 328 – L 333

Reviewer 3 Report

The author presented a revision to previous comments. There are still some points which must be improved. Authors must work on the following points:

 * The significance of the study should be mentioned clearly in abstract section. Data should be incorporated into the abstract.

* Line no 94-97 needs rephrasing of the sentence.

* The discussion and interpretation of results does not clearly explain its impact on the literature and the field.

* Discussions need to be supported by the latest references and need to be explained in depth. The authors should highlight the reason of their result findings in the light of available literature.

* It is strongly recommended to add a subsection, ‘practical implications of this literature,’ outlining the challenges in the current research, future work, and recommendations, before the conclusion.

* Conclusions is not just about summarizing the key results of the study, it should highlight the insights and the applicability of your findings/results for further work.  Please enrich your conclusion

* Check and correct grammatical and space errors throughout the article.

Author Response

Reviewer 3

The significance of the study should be mentioned clearly in abstract section. Data should be incorporated into the abstract.

We have completely rewritten the abstract emphasizing the importance of this study.

* Line no 94-97 needs rephrasing of the sentence.

The change suggested by the reviewer has been made.

* The discussion and interpretation of results does not clearly explain its impact on the literature and the field.

* Discussions need to be supported by the latest references and need to be explained in depth. The authors should highlight the reason of their result findings in the light of available literature.

* Discussions need to be supported by the latest references and need to be explained in depth. The authors should highlight the reason of their result findings in the light of available literature.

* It is strongly recommended to add a subsection, ‘practical implications of this literature,’ outlining the challenges in the current research, future work, and recommendations, before the conclusion.

* Conclusions is not just about summarizing the key results of the study, it should highlight the insights and the applicability of your findings/results for further work.  Please enrich your conclusion

A new subsection called "future challenges" has been created to show the latest trends on the problems of installing PRBs.

Round 2

Reviewer 3 Report

The authors have addressed all the comments, therefore the manuscript may be accepted in the present form. 

This manuscript is a resubmission of an earlier submission. The following is a list of the peer review reports and author responses from that submission.

Round 1

Reviewer 1 Report

Dear Editor,

Regarding the manuscript in question, I have read it and I think it is an interesting piece of work. 

I invite the authors reducing some sections. The paper is quite larger.

Minor suggestions:

  • Title  : instead ;
  • L45: Do not use "etc" in scientific papers
  • L236: Does Eq 9 is the linearized form of Eq 8? I think it is "linearized" form of other equation. It is not clear
  • Eq 10 is too obvious. Please remove it
  • Eq 15 and 16: Are the units of Langmuir and Freundlich coefficents "k" the same? (L3/M-1). Please check it.
  •  Table 1 and all text: standardize units: "or mg/L or mg L-1".
  • Figures must be improved (definition, quality)
  • Tables must be improved (broken lines and columns)
  • Table 4: second line appears to be wrong (R2=8).
  • Table 4: Please use RMSE instead of R2 in order to compare different nonlinear regressions.

Sincerely yours.

Reviewer 2 Report

Please, see the file attached

Reviewer 3 Report

This manuscript investigated the reactivity of different natural and synthetic materials towards three transition metals namely zinc, copper and cadmium. The study confirmed the significant adsorption capacity of the tested reagents against all transition metals. The manuscript has been written well and supported with sufficient experimental data. Please consider the following comments to improve the manuscript.

* The significance of the study should be mentioned clearly in abstract section. It should be rewritten by detailing the aim and concept of the paper. The abstract should state briefly the purpose of the research, the principal results and major conclusions.

* What is the current level of understanding in relation to removal of transition metals from contaminated aquifers by PRB Technology? What are the knowledge gaps?. These should be included in the introduction section. The introduction is insufficient to provide the state of the art in the topic. Hypothesis should be given. How this work is different from the available data?

The originality and novelty of the paper need to be further clarified. What progress against the most recent state-of-the-art similar studies was made in this study?

*The introduction of the manuscript must be extended and reformulated in order to provide a more comprehensive approach.

* The last paragraph or closing lines of the introduction section (objectives) needs further revision.

*Authors are suggested to add discussion by explaining trends in the obtained results along with the possible mechanisms behind the trends

*The manuscript does not provide interesting and technically sound discussion; it would be better to use more recent references in discussion.

*Under section discussion, it is recommended to discuss and explain what should be the appropriate policies based on the findings of this study. Also, the results should be further elaborated to show how they could be used for real applications. 

*Conclude with more focus on the major outcomes of the research work.